# TAFRO Syndrome: Guidance for Managing Patients Presenting Thrombocytopenia, Anasarca, Fever, Reticulin Fibrosis, Renal Insufficiency, and Organomegaly

**DOI:** 10.3390/biomedicines12061277

**Published:** 2024-06-08

**Authors:** Katsuhiro Miura, Haruna Nishimaki-Watanabe, Hiromichi Takahashi, Masaru Nakagawa, Shimon Otake, Takashi Hamada, Takashi Koike, Kazuhide Iizuka, Yuuichi Takeuchi, Kazuya Kurihara, Toshihide Endo, Shun Ito, Hironao Nukariya, Takahiro Namiki, Yoshiyuki Hayashi, Hideki Nakamura

**Affiliations:** 1Department of Medicine, Division of Hematology and Rheumatology, Nihon University School of Medicine, 30-1 Oyaguchikamicho, Itabashi City, Tokyo 173-8610, Japan; takahashi.hiromichi@nihon-u.ac.jp (H.T.); nakagawa.masaru@nihon-u.ac.jp (M.N.); ootake.shimon@nihon-u.ac.jp (S.O.); hamada.takashi@nihon-u.ac.jp (T.H.); koike.takashi@nihon-u.ac.jp (T.K.); iizuka.kazuhide@nihon-u.ac.jp (K.I.); takeuchi.yuuichi@nihon-u.ac.jp (Y.T.); kurihara.kazuya@nihon-u.ac.jp (K.K.); endo.toshihide@nihon-u.ac.jp (T.E.); ito.shun@nihon-u.ac.jp (S.I.); nukariya.hironao@nihon-u.ac.jp (H.N.); namiki.takahiro75@nihon-u.ac.jp (T.N.); hayashi.yoshiyuki@nihon-u.ac.jp (Y.H.); nakamura.hideki@nihon-u.ac.jp (H.N.); 2Department of Pathology and Microbiology, Division of Oncologic Pathology, Nihon University School of Medicine, 30-1 Oyaguchikamicho, Itabashi City, Tokyo 173-8610, Japan; nishimaki.haruna@nihon-u.ac.jp; 3Department of Pathology and Microbiology, Division of Laboratory Medicine, Nihon University School of Medicine, 30-1 Oyaguchikamicho, Itabashi City, Tokyo 173-8610, Japan

**Keywords:** idiopathic multicentric Castleman disease, interleukin-6, siltuximab, TAFRO syndrome, tocilizumab

## Abstract

TAFRO syndrome is an inflammatory disorder of unknown etiology characterized by thrombocytopenia, anasarca, fever, reticulin fibrosis, renal insufficiency, and organomegaly. Despite great advancements in research on the TAFRO syndrome in the last decade, its diagnosis and treatment are still challenging for most clinicians because of its rarity and severity. Since the initial proposal of the TAFRO syndrome as a distinct disease entity in 2010, two independent diagnostic criteria have been developed. Although these are different in the concept of whether TAFRO syndrome is a subtype of idiopathic multicentric Castleman disease or not, they are similar except for the magnitude of lymph node histopathology. Because there have been no specific biomarkers, numerous diseases must be ruled out before the diagnosis of TAFRO syndrome is made. The standard of care has not been fully established, but interleukin-6 blockade therapy with siltuximab or tocilizumab and anti-inflammatory therapy with high-dose corticosteroids are the most commonly applied for the treatment of TAFRO syndrome. The other immune suppressive agents or combination cytotoxic chemotherapies are considered for patients who do not respond to the initial treatment. Whereas glowing awareness of this disease improves the clinical outcomes of patients with TAFRO syndrome, further worldwide collaborations are warranted.

## 1. Introduction

TAFRO syndrome is a systemic inflammatory disease characterized by *t*hrombocytopenia, *a*nasarca (generalized edema), *f*ever, *r*eticulin fibrosis (myelofibrosis), *r*enal insufficiency, and *o*rganomegaly (hepatosplenomegaly and lymphadenopathy) [1]. The disease usually progresses rapidly, and most patients with TAFRO syndrome are critically ill. However, the pathogenesis of the TAFRO syndrome remains largely unknown owing to its rarity and novelty. Although the differential diagnoses of patients with these symptoms vary widely, no specific biomarkers have been identified. Thus, treating the TAFRO syndrome remains challenging for most physicians. 

The goal of this review was to understand the disease concept, diagnostic procedures, and current treatment paradigms for TAFRO syndrome. To better understand this complex disease, we begin with the definition of TAFRO syndrome as it is recognized today. Although numerous sporadic cases have been reported, our focus is exclusively on published diagnostic criteria, studies utilizing registry databases, and reports of case series to concisely illustrate the disease characteristics of TAFRO syndrome. Furthermore, we have excluded in vivo studies and experimental therapies that are not widely accepted. The relationship between TAFRO syndrome and idiopathic multicentric Castleman disease (iMCD) is highlighted as the key concept to enhance our understanding of TAFRO syndrome.

## 2. What Is TAFRO Syndrome?

### 2.1. Establishment of Disease Concept

The TAFRO syndrome was first proposed in 2010 by Takai, a Japanese hematologist, and colleagues as a unique disease concept that could not be classified into any existing clinical entities based on their three cases [1]. Subsequently, researchers, mainly in Japan, reported a series of similar cases, and data on the clinical picture and pathological findings of patients with TAFRO syndrome have accumulated [2,3,4,5,6]. According to their reports, the pathological findings of lymph node biopsies in TAFRO syndrome cases resembled those observed in Castleman disease, specifically the mixed or hyaline-vascular types. At that time, researchers considered these cases to be identical to the non-idiopathic plasmacytic lymphadenopathy (non-IPL) type of iMCD, which was previously reported by Kojima et al. in 2008, detailing the histopathological findings from iMCD cases complicated by pleural effusions and thrombocytopenia [7]. For instance, Japanese researchers used the term “Castleman-Kojima disease” in the titles of early articles [6]. The above discoveries have brought the TAFRO syndrome to researchers’ international attention. However, many unresolved issues remain, such as the frequency, etiology, optimal treatment, and prognosis of TAFRO syndrome. Therefore, there is an urgent need to establish diagnostic criteria and treatment guidelines for this syndrome.

### 2.2. Development of Diagnostic Criteria

Currently, there are two independent diagnostic criteria for TAFRO syndrome, which differ in how they compare to Castleman’s disease. Accordingly, physicians should be familiar with both diagnostic criteria and compare them appropriately to diagnose TAFRO syndrome.

Iwaki, in collaboration with Fajgenbaum et al., reported 23 cases from Japan and two cases from the U.S., the largest number at that time, with detailed clinical symptoms and pathological findings of lymph nodes from patients with TAFRO syndrome [8]. They advocated that TAFRO syndrome was a subtype of iMCD and demonstrated “Proposed Diagnostic Criteria for TAFRO-iMCD”, published online in November 2015 [8]. The criteria were later revised to the international definition of “iMCD-TAFRO” in 2021 [9]. Because Castleman disease was developed with an emphasis on pathological diagnosis, the diagnosis of TAFRO-iMCD mandates pathological diagnosis by lymph node biopsy. This strict definition allows for a more homogeneous diagnosis but simultaneously prevents the diagnosis of severely ill patients who cannot undergo lymph node biopsy. 

Meanwhile, the Japanese TAFRO Syndrome Research Team, led by Masaki et al., developed the “Proposed diagnostic criteria, disease severity classification, and treatment strategy for TAFRO syndrome, 2015 version” based on a consensus meeting held on 28 case studies [10]. The authors concluded that TAFRO syndrome is distinct from iMCD, although they share some clinical and pathological features. These criteria were initially published online in March 2016 and updated in 2019 [10,11]. Notably, they do not require a histopathological diagnosis because some patients cannot undergo a lymph node biopsy owing to a severe bleeding tendency or a lack of enlarged lymph nodes in the initial presentation in approximately 30% of patients. While this definition can be used to diagnose severely ill patients in whom lymph node biopsy is impossible, a heterogeneous group of diseases may be diagnosed as TAFRO syndrome owing to the lack of pathological evidence [12]. For this reason, the authors highlighted that lymph node biopsy “is important and strongly recommended” in the “Disease description” section at the beginning of the diagnostic criteria.

### 2.3. Relevance to Castleman’s Disease

Although both TAFRO syndrome and iMCD are forms of cytokine storms largely dependent on interleukin-6 (IL-6) upregulation or the secretion of other cytokines, such as vascular endothelial growth factor (VEGF) [13,14], the debate over how TAFRO syndrome should be positioned in relation to iMCD is yet to be definitively answered. According to the international consensus diagnostic criteria for iMCD published by Fajgenbaum et al., patients with lymph node biopsy findings consistent with iMCD and clinical signs of TAFRO syndrome are classified as “iMCD patients with TAFRO syndrome” or “iMCD cases with TAFRO clinical features”. These are considered subtypes or severe forms of iMCD [15,16]. Conversely, some experts believe TAFRO syndrome is distinct from Castleman disease because the clinical features and prognosis of patients with TAFRO syndrome significantly differ from those of patients with iMCD without TAFRO syndrome [17,18].

Owing to the decade of controversies mentioned above, understanding TAFRO syndrome remains complicated for clinicians and researchers. For instance, the authors of the international definition of iMCD-TAFRO further divided TAFRO syndrome into three categories: “iMCD-TAFRO”, “TAFRO with probable iMCD”, and “TAFRO without iMCD and other co-morbidities” [9]. However, such terminology requires further investigation and careful discussion for its validity to progress in research on the pathophysiology of TAFRO syndrome [19].

## 3. How to Diagnose TAFRO Syndrome

### 3.1. Clinical Application of Diagnostic Criteria

A comparison of the international definition of iMCD-TAFRO and the 2019 updated criteria of the Japanese TAFRO Syndrome Research Team is shown in Table 1. These two diagnostic criteria do not differ greatly except for the magnitude of lymph node histopathology. They require major symptoms such as thrombocytopenia, generalized edema and fluid retention, fever, myelofibrosis with increased bone marrow megakaryocytes, hepatosplenomegaly, lymphadenopathy (organ enlargement), renal involvement, and liver-derived alkaline phosphatase elevation. Both criteria require the exclusion of various diseases that may cause similar symptoms. We recommend the use of both criteria, especially in cases where a lymph node biopsy has been performed. If lymph node biopsy is not possible or contraindicated, the “TAFRO Syndrome Study Group Diagnostic Criteria” should be used.

A wide range of diseases must be excluded when diagnosing TAFRO syndrome because dozens of infectious, autoimmune, and cancerous diseases (e.g., tuberculosis, sarcoidosis, systemic lupus erythematosus, vasculitis syndrome, Sjogren’s syndrome, T-cell lymphomas, intravascular large B-cell lymphoma, and other solid cancers) can exhibit TAFRO-like symptoms [20]. Furthermore, no highly specific and reliable biomarkers have yet been identified; the diagnosis of TAFRO syndrome is based on a meticulous differential diagnosis. The 2019 updated criteria from the Japanese TAFRO Syndrome Research Team detail the procedures for excluding other diseases (cf. “Point to consider”, the right column of the “Supportive items” in Table 1). For example, whole-blood interferon-gamma release assays and adenosine deaminase measurements in pleural effusions are recommended to rule out tuberculosis. A random skin biopsy is also suggested when intravascular large B-cell lymphoma is suspected. However, in the 2019 updated version, the authors removed IgG4-related diseases (IgG4-RD) from the list of “Diseases to be excluded” to avoid the misinterpretation that IgG4-RD can cause symptoms such as TAFRO syndrome. For clinicians familiar with IgG4-RD, differentiating IgG4-RD from TAFRO syndrome is not challenging because systemic inflammation with fever exceeding 38 °C and serum c-reactive protein (CRP) elevation is generally absent in IgG4-RD [21,22]. This is in contrast to the differential diagnosis between IgG4-DR and iMCD-ILD, which is sometimes challenging even for skilled practitioners [23].

Regardless of whether TAFRO syndrome is a part of iMCD, one of the most important clinical differences between TAFRO syndrome and iMCD-TAFRO versus iMCD without TAFRO is the clinical course. TAFRO syndrome has an acute or subacute onset, and the patient’s general condition usually deteriorates quickly, whereas most iMCD cases have a relatively chronic course. Thus, life-threatening multi-organ involvement and hospitalization burdens are more common in TAFRO syndrome [24]. Another aspect that differentiates these two conditions is that polyclonal hypergammaglobulinemia, as definitively seen in iMCD without TAFRO, is rarely observed in TAFRO syndrome. However, despite its standpoint on TAFRO syndrome and iMCD, the Japanese TAFRO Syndrome Research Team criteria do not list iMCD as “Diseases to be excluded” from TAFRO syndrome because a subset of patients with TAFRO syndrome meets the criteria for iMCD [11]. Similarly, immune thrombocytopenic purpura was omitted from the “Diseases to be excluded” because the underlying cause of thrombocytopenia may be autoimmunity [25]. It is important to note that autoantibodies, such as antinuclear antibodies, often test positive in this disease. Skilled hematologists and rheumatologists should collaborate on the differential diagnosis of TAFRO syndrome.

### 3.2. Histopathological Diagnosis of Lymph Nodes

A lymph node biopsy should be attempted whenever possible if a patient is suspected of having TAFRO syndrome. Because several diseases display TAFRO syndrome-like symptoms, the diagnosis of TAFRO syndrome is ideally made after excluding these mimickers by histopathology of the lymph node. Because enlarged lymph nodes can reach up to 15 mm in diameter in cases of TAFRO syndrome [8,10], lymph nodes larger than this may suggest other lymphoproliferative disorders. It is also crucial to not overlook malignant diseases, even with only mild enlargement of the lymph nodes. Forgoing a lymph node biopsy is justified only in situations where no enlarged lymph nodes are confirmed or the patient cannot tolerate an invasive biopsy due to a bleeding tendency from thrombocytopenia or significant systemic edema.

The original Proposed Diagnostic Criteria for TAFRO-iMCD by Iwaki et al., which mandates histopathological findings in lymph nodes, defined pathological characteristics of TAFRO-iMCD as “atrophic germinal centers with enlarged nuclei of endothelial cells, the proliferation of endothelial venules with enlarged nuclear in the interfollicular zone, and small numbers of mature plasma cells [8]”. This finding indeed resembled the formerly called “hyaline-vascular” subtype of iMCD. However, in TAFRO-iMCD, the lymph node specimen does not show the typical hyaline-vascular structures aggressively penetrating the germinal center, which is usually seen in unicentric Castleman’s disease (UCD) [26]. In the international, evidence-based consensus diagnostic criteria for HHV-8–negative/idiopathic MCD, such pathological findings are referred to as the “hypervascular” histopathologic subtype, typically seen in TAFRO-iMCD but occasionally found in iMCD without TAFRO syndrome [15]. Recently, the histopathology of iMCD has been recognized as having a gradation from the hypervascular spectrum to the plasmacytic spectrum rather than the three strictly classifiable categories of hypervascular, mixed, or plasmacytic subtypes [27,28]. Following this concept, the newly revised international definition of iMCD-TAFRO, which also mandates a lymph node biopsy, states that the lymph nodes “must be consistent with the histopathologic features of the International iMCD Diagnostic Criteria”. This includes, in brief, atrophic germinal centers, concentric rings of mantle zone cells, and interfollicular hypervascularization. The criteria specify that samples must test negative for light chain restriction and HHV-8 [9].

Conversely, the Japanese TAFRO Syndrome Research Team criteria do not require the acquisition of lymph node histopathology; instead, they categorize it as a minor criterion, described as “Castleman disease-like features on lymph node biopsy”. This is because the primary purpose of conducting a lymph node biopsy is to rule out malignant lymphoma or other diseases. Kurose, a coauthor of the Japanese TAFRO Syndrome Research Team’s criteria, et al. conducted a quantitative analysis comparing the pathological findings in lymph nodes of iMCD cases with (*n* = 37) or without (*n* = 33) TAFRO syndrome according to these diagnostic criteria [29]. They reported that iMCD with TAFRO showed significantly more atrophic lymphoid follicles, greater distances between follicles, and increased glomerular vascular proliferation within the germinal center compared to iMCD without TAFRO. Particularly, all “hypervascular” subtype cases (*n* = 6) were found only in iMCD with TAFRO, demonstrating severe atrophic lymphoid follicles and interfollicular vascular proliferation.

Altogether, the presence of hypervascular subtype features in lymph node pathology, potentially reflecting VEGF secretion [30], is more likely to lead to the diagnosis of TAFRO syndrome (Figure 1). It is important to note that the histological features of the “hypervascular” subtype in iMCD-TAFRO and the “hyaline-vascular” subtype in UCD are distinct entities with differing characteristics. The presence or absence of light chain restriction on immunohistochemical staining is crucial for distinguishing it from plasma cell neoplasms; however, marked plasma cell proliferation is rare in TAFRO syndrome. Therefore, the plasmacytic subtype histology found in patients with suspected TAFRO syndrome requires careful consideration of other potential autoimmune diseases [31]. This preference for hypervascular histology of TAFRO syndrome can be contrasted with that of iMCD-IPL, which principally harbors plasmacytic pathology (cf., “non-IPL type of iMCD”). Recently, iMCD-IPL has been considered a distinct subtype of iMCD, characterized by an indolent clinical course, favorable response to treatment, and excellent overall survival [32,33,34,35].

### 3.3. Other Findings

Bone marrow pathology is not unique to TAFRO syndrome; however, a bone marrow biopsy should be performed to differentiate various hematopoietic diseases unless contraindicated. Typically, bone marrow aspiration results in a dry tap, and the biopsy shows myelofibrosis and megakaryocytic hyperplasia with atypia [36].

While up to half of the TAFRO syndrome cases present renal insufficiency, few patients can undergo renal biopsy due to the significant risk of serious bleeding. In a small case series analyzing seven patients with TAFRO syndrome, all demonstrated “glomerular endotheliopathy characterized by endothelial cell swelling and a double contour of the glomerular basement membrane with mesangiolysis or mesangial loosening [37]”. The specificity of this finding or its relationship with other possible mechanisms, such as thrombotic microangiopathy, remains unknown [38]. However, renal biopsy may be considered particularly for patients who need to exclude other diseases possibly complicated by renal damage, such as vasculitis with or without antineutrophil cytoplasmic antibodies.

Despite the excessive symptoms of anasarca, reticulin fibrosis of the bone marrow, and organomegaly, the findings in most imaging studies are uncharacteristic of patients with TAFRO syndrome. Kizaki et al. reported that chest computed tomography identified a “matted” appearance of the enlarged anterior mediastinum without solid mass, explained by the mixture of fat and soft tissue densities, in 7 out of 11 patients with TAFRO syndrome [39]. This imaging feature may help in the diagnosis of TAFRO syndrome, particularly when critical findings (e.g., lymph node histology) are not available (Figure 2).

## 4. How to Treat TAFRO Syndrome

Once TAFRO syndrome is diagnosed, treatment should be initiated immediately. Patients with TAFRO syndrome are treated empirically with high-dose corticosteroids, monoclonal antibodies against IL-6 (siltuximab) or its receptor IL-6R (tocilizumab), the anti-CD20 monoclonal antibody rituximab, calcineurin inhibitors such as cyclosporine, cytotoxic chemotherapies (e.g., R-CHOP, rituximab, cyclophosphamide, doxorubicin, vincristine, and prednisone), or other immunosuppressive medications, either as monotherapy or in combination. These treatment options for TAFRO syndrome are based exclusively on expert opinions, the experiences of skilled physicians, case reports, or retrospective analyses of registry data, as no conclusive evidence from randomized studies exists to date (Table 2).

The international evidence-based consensus treatment guidelines for iMCD do not distinguish treatment strategies for iMCD with or without TAFRO syndrome; that is, IL-6-blockade with siltuximab or tocilizumab is the standard of care for these populations [40]. Additionally, they recommend the concurrent use of high-dose corticosteroids with initial treatment using anti-IL-6(R) antibodies and suggest considering more aggressive regimens, such as R-CHOP, for refractory cases in “Severe iMCD”, where most cases of TAFRO syndrome are categorized [28,40]. According to treatment data from 102 patients with iMCD, including 60 patients who met the TAFRO criteria in the ACCELERATE Natural History Registry, Pierson et al. reported that 10 of 21 (47.6%) and 5 of 13 patients with iMCD and TAFRO (38.5%) responded to treatment with siltuximab and tocilizumab with or without corticosteroids, respectively, although none of the 22 patients (0%) responded to treatment with corticosteroids alone [41]. Of the 23 patients with the TAFRO subtype who underwent cytotoxic chemotherapy, 11 (47.8%) responded to treatment [41].

In “The 2015 treatment strategy for TAFRO syndrome”, published by the Japanese TAFRO Syndrome Research Team and not updated in the 2019 version, high-dose glucocorticoid, i.e., oral prednisolone at 1 mg/kg/day for 2 weeks, followed by tapering with or without methyl-prednisolone pulse therapy at 500–1000 mg/day for 3 days, is recommended for first-line therapy [11]. Although this recommendation was based on observations that some patients with TAFRO syndrome were successfully treated with corticosteroids alone, most required additional therapies. Consequently, cyclosporine, tocilizumab, or rituximab are recommended as second-line therapies, along with thrombopoietin receptor agonists for persistent thrombocytopenia [10]. To further explore the optimal treatment for TAFRO syndrome, they detailed the clinical outcomes of 81 patients in their registry, the Multicenter Collaborative Retrospective Study for Establishing the Concept of TAFRO Syndrome Registry (UMIN000011809) [42]. Fujimoto et al. reported that 68 of 81 patients with TAFRO syndrome were treated with first-line therapy with corticosteroids and analyzed 43 of 68 patients who subsequently received second-line therapy with tocilizumab (*n* = 21), cyclosporine (*n* = 14), or rituximab (*n* = 8). Accordingly, landmark analysis from the time of second-line treatment initiation revealed that the median time to the next treatment varied among the three regimens (2.8, 9.2 months, and not reached for the tocilizumab, cyclosporine, and rituximab groups, respectively; *p* = 0.030), while overall survival was not significantly different [42]. In a subset analysis of 21 patients who received corticosteroid therapy alone because of either the unnecessary need for additional therapy or death, the 1-year overall survival rate was 71.4% [42]. Using the same registry, including 83 patients with TAFRO syndrome, Kawabata et al. demonstrated that patients’ ages of ≥60 years and elevated plasma D-dimer levels of ≥18 μg/dL could be adverse prognostic factors [43].

Based on these data, the current treatment for TAFRO syndrome consists of corticosteroids and IL-6/IL-6R blockade with siltuximab or tocilizumab. Other immunosuppressive or cytotoxic treatments may be an additional treatment choice depending on the response to first-line therapy, the patient’s condition, the drug approval status in each region, or the physician’s preference. Supportive treatment with oxygen supplementation, fluid infusion management, infection control, transfusion, nutrition, and rehabilitation plays vital roles in the management of TAFRO syndrome. Tentative hemodialysis or mechanical ventilation is occasionally required until the patient’s recovery or treatment response. Our recent treatment approach is summarized in Figure 3.

**Table 2 biomedicines-12-01277-t002:** Results of published studies of different therapeutic modalities for TAFRO syndrome.

References	Agents	Clinical Outcomes *
Iwaki et al. [8]	Corticosteroids alone (1st line) Additional therapy with cyclosporine, tocilizumab, rituximab, or others	11 of 23 (47.8%) patients responded 9 of 12 (75%) patients responded
Pierson et al. [41]	Siltuximab ± corticosteroids Tocilizumab ± corticosteroids Corticosteroids alone Rituximab Chemotherapy-based Other	10 of 21 (47.6%) patients responded 5 of 13 (38.5%) patients responded None of the 22 patients responded 3 of 10 (30%) patients responded 11 of 23 (47.8%) patients responded 14 of 31 (45.2%) patients responded
Fujimoto et al. [42]	Corticosteroids alone (1st line) Tocilizumab (2nd line) Cyclosporine (2nd line) Rituximab (2nd line)	1-year survival rate of 71.4% (*n* = 21) Median time to next treatment (TTNT) of 2.8 months 1-year survival rate of 71.4% (*n* = 21) Median TTNT of 9.2 months 1-year survival rate of 64.3% (*n* = 14) Median TTNT not reached 1-year survival rate of 87.5% (*n* = 8)
Akiyama et al. [44]	Tocilizumab + corticosteroids or various agents	A total 31 patients (1st line, *n* = 18 and later line, *n* = 13) 16 (51.6%) achieved complete response, and 15 patients showed only partial or no response

* Definition of the response was not uniform among the studies.

## 5. Conclusions

Owing to the continuing efforts of researchers over the past decade, global awareness of TAFRO syndrome has recently increased [45]. Its disease entity is now generally accepted, and its treatment has been standardized over the last few years.

Nevertheless, a substantial proportion of patients with TAFRO syndrome do not respond to IL-6-blockade therapy concomitantly with corticosteroids, which is now the most commonly selected initial treatment [44]. Additionally, the clinical behavior or response to treatment can vary geographically according to the current diagnostic criteria or recommendations [46]. Thus, to enhance the clinical outcomes of patients with this extremely rare and overlooked disease, international collaborations and unified efforts to establish diagnostic criteria and treatment guidelines for TAFRO syndrome are eagerly anticipated.

## Figures and Tables

**Figure 1 biomedicines-12-01277-f001:**
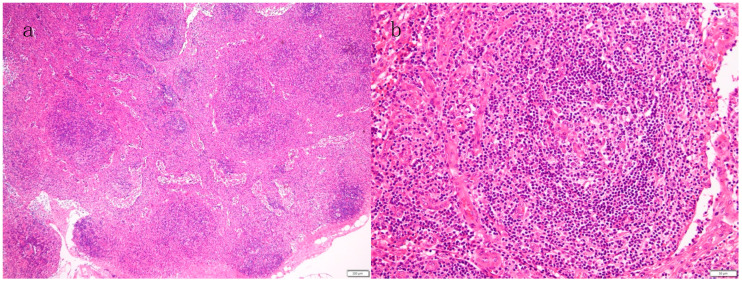
A representative case with thrombocytopenia, anasarca, fever, reticulin fibrosis, and organomegaly (TAFRO syndrome). The histopathology of the inguinal lymph node specimen showed atrophic or vanishing germinal centers (**a**, H & E). Hypervascularization with endothelial cell proliferation is seen in the germinal center and interfollicular area (**b**, H & E).

**Figure 2 biomedicines-12-01277-f002:**
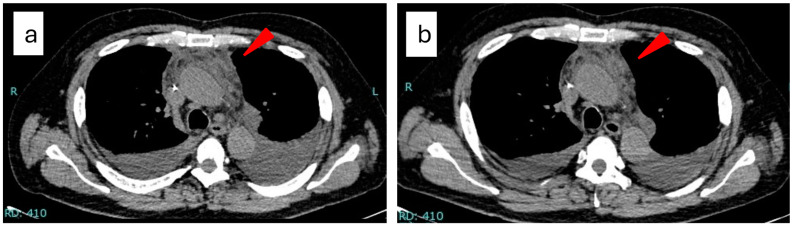
A man in his sixties presented with thrombocytopenia, pleural effusion, ascites, fever, renal insufficiency, and mild splenomegaly. The patient exhibited no enlarged superficial lymph nodes. A bone marrow biopsy showed reticulin fibrosis and megakaryocyte hyperplasia, which are compatible with the findings of TAFRO syndrome. A random skin biopsy was negative for intravascular large B-cell lymphoma. The chest computed tomography demonstrated a “matted” appearance of the enlarged anterior mediastinum (**a**, red arrow). The lesion slightly regressed 1 week after the initiation of the treatment with 1 mg/kg of prednisolone (**b**, red arrow).

**Figure 3 biomedicines-12-01277-f003:**
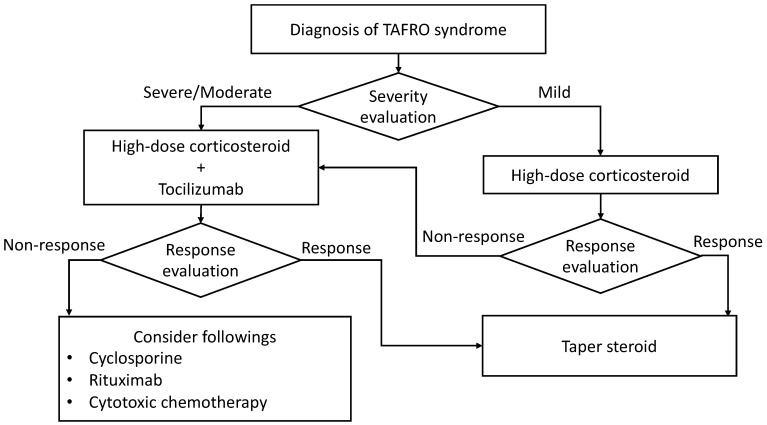
Our current treatment algorithm for patients with TAFRO syndrome. Disease severity is evaluated according to the 2019 updated diagnostic criteria and disease severity classification for TAFRO syndrome.

**Table 1 biomedicines-12-01277-t001:** Comparison of diagnostic criteria for iMCD-TAFRO and TAFRO syndrome.

	International Definition of iMCD-TAFRO	2019 Updated Criteria for TAFRO Syndrome
Histopathology	Mandatory Lymph node must be consistent with histopathologic features of the International iMCD Diagnostic Criteria	Categorized in minor item (see below); however, as it is very important to exclude malignancies including lymphoma, lymph node biopsy is strongly recommended
Major items	All four required Thrombocytopenia (≤100 × 10^3^/μL) Anasarca Fever (≥37.5 °C) or hyperinflammatory (CRP ≥ 2 mg/dL) status Organomegaly (small volume lymphadenopathy in ≥2 regions, hepatomegaly, or splenomegaly)	All three required Anasarca Thrombocytopenia (≤100 × 10^3^/μL) Systemic inflammation defined as fever ≥ 37.5 °C and/or CRP ≥ 2 mg/dL
Minor items	At least one required Renal insufficiency (eGFR ≤ 60 mL/min/1.73 m^2^, creatinine > 1 mg/dL for females and 1.3 mg/dL for males, or renal failure necessitating hemodialysis) TAFRO-consistent bone marrow (reticulin fibrosis or megakaryocytic hyperplasia without evidence of an alternative diagnosis)	At least two required Castleman disease-like features on lymph node biopsy Reticulin myelofibrosis and/or increased number of megakaryocytes in bone marrow Mild organomegaly (hepatomegaly, splenomegaly, and lymphadenopathy) Progressive renal insufficiency
Supportive items	Not required but strongly supportive Absence of polyclonal hypergammaglobulinemia (IgG ≤ 1.2 × normal upper limit) Elevated ALP with mild to no elevation in bilirubin and transaminases	Points to consider Marked polyclonal hypergammopathy (IgG > 3000 mg/dL) is rare in TAFRO Obvious monoclonal protein should be absent Few patients show elevated serum LDH, which indicates lymphoma. Especially intravascular large B-cell lymphoma mimics TAFRO syndrome, and random skin biopsy is recommended in such cases. Most patients show elevated levels of serum ALP Hepatosplenomegaly is usually mild and only confirmed by CT-scan Lymphadenopathy in this disease is usually smaller than 1.5 cm in diameter Exclusion criteria for Castleman disease and ITP have not been determined To exclude autoimmune disorders, rheumatoid factor, anti-nuclear antibody, anti-SS-A/Ro antibody, MPO-ANCA, and PR3-ANCA have to be examined To exclude of mycobacterial infections, examination of interferon-gamma release assays and ADA in pleural effusion is recommended Pleural effusion and ascites in patients are usually transudative, but concentrations of IL-6 and VEGF in those fluids are usually higher than those in serum
Exclusions	Must rule out the following diseases: Infectious diseases—including the below but not limited to 1. HHV-8 2. EBV-associated 3. Lymphoproliferative disorders 4. Acute HIV infection 5. Tuberculosis 6. COVID-19 cytokine storm syndrome Autoimmune/rheumatologic diseases: 1. Systemic lupus erythematosus 2. Sjögren syndrome 3. Rheumatoid arthritis 4. Adult-onset Still disease 5. Juvenile idiopathic arthritis 6. IgG ≥ 3400 mg/dL (suggestive of autoimmune diseases or plasma cell dyscrasias) 7. Primary hemophagocytic lymphohistiocytosis Malignancy—including the below but not limited to the following: 1. Malignant lymphoma 2. Multiple myeloma 3. Metastatic cancer 4. POEMS syndrome	Diseases to be excluded: Malignancies, including lymphoma, myeloma, and mesothelioma, etc. Autoimmune disorders, including systemic lupus erythematosus, Sjogren’s syndrome, and ANCA-associated vasculitis, etc. Infectious disorders, including acid-fast bacterial infection, rickettsial disease, Lyme disease, severe fever with thrombocytopenia syndrome, etc. POEMS syndrome Hepatic cirrhosis Thrombotic thrombocytopenic purpura/ hemolytic uremic syndrome

iMCD, idiopathic multicentric Castleman disease; CRP, C-reactive protein; eGFR, estimated glomerular filtration rate; IgG, immunoglobulin G; LDH, lactate dehydrogenase; ALP, alkaline phosphatase; CT, computed tomography; MPO, myeloperoxidase; ANCA, anti-neutrophil cytoplasmic antibody; PR3, proteinase 3; ADA, adenosine deaminase; IL-6, interleukin-6; VEGF, vascular endothelial growth factor; ITP, immune thrombocytic purpura; HHV-8, human herpesvirus-8; EBV, Epstein–Barr virus; HIV, human immunodeficiency virus.

## Data Availability

Data in this article are available upon reasonable request from the corresponding author.

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
