# Peer review of "TAFRO Syndrome: Guidance for Managing Patients Presenting Thrombocytopenia, Anasarca, Fever, Reticulin Fibrosis, Renal Insufficiency, and Organomegaly"

_biomedicines, 2024, doi:10.3390/biomedicines12061277_

Round 1
Reviewer 1 Report
Comments and Suggestions for Authors
A very interesting paper dedicated to the diagnosis and therapy of a very rare and neglected disorder. The paper is an adequate presentation of existing knowledge. However, better systematization and presentation is needed. I suggest that the authors:
1. Add a table with differential diagnosis and recommended diagnostic procedures
2. Add photos with a typical pathohistological finding
3. Add a table with the results of studies published so far with different therapeutic modalities
4. After summarizing all the knowledge published so far, it would be useful for the authors to propose their own treatment algorithm
Author Response
We are grateful to the reviewers for their effort and time spent in meticulously reviewing our manuscript and for their valuable comments and suggestions. We have diligently addressed all the comments. Below are the reviewers' comments, followed by our responses. We believe that the revised manuscript is now suitable for publication in Biomedicines.
Revision Notes for Referee #1
A very interesting paper dedicated to the diagnosis and therapy of a very rare and neglected disorder. The paper is an adequate presentation of existing knowledge. However, better systematization and presentation is needed. I suggest that the authors:
- Add a table with differential diagnosis and recommended diagnostic procedures
The details of the differential diagnosis and recommended diagnostic procedures are listed in "Point to consider," the right column of the "Supportive items" in Table 1. To draw the reader's attention, we have inserted parentheses as follows: "The 2019 updated criteria from the Japanese TAFRO Syndrome Research Team detail the procedures for excluding other diseases (cf. "Point to consider," the right column of the "Supportive items" in Table 1)." (Lines 141–143).
- Add photos with a typical pathohistological finding
We have added a figure displaying representative pathological findings of TAFRO syndrome (Line 235–240). Moreover, we have included a figure of a representative case exhibiting a "matted pattern" in the chest computed tomography (Line 264–271). These figures were initially part of the submitted manuscript but were requested to be removed by an editor as the original data should not be included in the review article. Nevertheless, we contend that these figures significantly augment the reader’s comprehension of TAFRO syndrome, aligning with the ultimate goal of this article. Consequently, we have updated the Author Contributions, Institutional Review Board Statement, Informed Consent Statement, and Data Availability Statement sections (Line 354–367).
- Add a table with the results of studies published so far with different therapeutic modalities
We have included a table that summarizes the results from published studies on various therapeutic modalities for TAFRO syndrome (Table 2) (Line 335–337).
- After summarizing all the knowledge published so far, it would be useful for the authors to propose their own treatment algorithm
We have included a figure that summarizes our current treatment algorithm (Fig 3).
Reviewer 2 Report
Comments and Suggestions for Authors
Very nicely written review of the definition, diagnosis and treatment options of TAFRO syndrome. Although, I personally think that TAFRO is subset of i-MDC, authors approached this topic with unbiased review of all options. Understandably treatment options review is the weakest part of the paper, since there is no hard data supporting specific options and we do not expect there will be randomized studies in this relatively rare disease. Therefore, options for the treatment will be chosen by treating physicians and we have to accept bias based on their personal preferences. I recommend authors to more strongly advocate in this paper international collaboration and attempts to coordinate effort to develop unified diagnostic criteria and guidelines for the treatment of TAFRO similar to those developed for i-MDC.
Author Response
Revision Notes for Referee #2
Very nicely written review of the definition, diagnosis and treatment options of TAFRO syndrome. Although, I personally think that TAFRO is subset of i-MDC, authors approached this topic with unbiased review of all options. Understandably treatment options review is the weakest part of the paper, since there is no hard data supporting specific options and we do not expect there will be randomized studies in this relatively rare disease. Therefore, options for the treatment will be chosen by treating physicians and we have to accept bias based on their personal preferences. I recommend authors to more strongly advocate in this paper international collaboration and attempts to coordinate effort to develop unified diagnostic criteria and guidelines for the treatment of TAFRO similar to those developed for i-MDC.
We appreciate the reviewer's favorable comments on our manuscript and concur with the reviewer's opinions. We have revised the sentences at the bottom of the abstract and the conclusions section as follows: "worldwide investigations" has been changed to "worldwide collaborations"; thus, to enhance the clinical outcomes of patients with this extremely rare and overlooked disease, international collaborations and unified efforts to establish diagnostic criteria and treatment guidelines for TAFRO syndrome are eagerly anticipated.